# Antioxidant Activity and Seasonal Variations in the Composition of Insoluble Fiber from the Cladodes of *Opuntia ficus-indica* (L.) Miller: Development of New Extraction Procedures to Improve Fiber Yield

**DOI:** 10.3390/plants13040544

**Published:** 2024-02-16

**Authors:** Rosamaria Caminiti, Maria Serra, Saverio Nucera, Stefano Ruga, Francesca Oppedisano, Federica Scarano, Roberta Macrì, Carolina Muscoli, Ernesto Palma, Vincenzo Musolino, Giancarlo Statti, Vincenzo Mollace, Jessica Maiuolo

**Affiliations:** 1IRC-FSH Center, Department of Health Sciences, University “Magna Græcia” of Catanzaro, Germaneto, 88100 Catanzaro, Italy; rosamariacaminiti4@gmail.com (R.C.); maria.serra@studenti.unicz.it (M.S.); rugast1@gmail.com (S.R.); oppedisanof@libero.it (F.O.); federicascar87@gmail.com (F.S.); robertamacri85@gmail.com (R.M.); palma@unicz.it (E.P.); mollace@libero.it (V.M.); 2Veterinary Pharmacology Laboratory, Institute of Research for Food Safety and Health (IRC-FSH), Department of Health Sciences, University “Magna Græcia” of Catanzaro, Germaneto, 88100 Catanzaro, Italy; 3Laboratoy of Pharmaceutical Biology, IRC-FSH Center, Department of Health Sciences, University “Magna Græcia” of Catanzaro, Germaneto, 88100 Catanzaro, Italy; v.musolino@unicz.it; 4Department of Pharmacy, Health and Nutritional Sciences, University of Calabria, 87100 Cosenza, Italy; giancarlo.statti@unical.it; 5Fondazione R. Dulbecco, 88046 Lamezia Terme, Italy

**Keywords:** *Opuntia ficus-indica*, cladodes, insoluble fiber, fiber extraction protocol, antioxidant properties, polyphenols

## Abstract

*Opuntia ficus-indica (L.)* Miller is a plant belonging to the *Cactaceae* family adapted to live in environments characterized by long periods of drought and arid or desert climates. This plant is characterized by an aerial part composed of structures transformed by branches, called “cladodes”, which are essential to reduce excessive perspiration of water and appear covered with thorns. The composition of the cladodes includes water, polysaccharides, fiber, proteins, vitamins, fatty acids, sterols, polyphenols, and minerals. The main purposes of this scientific work are (a) to compare the insoluble fiber (IF) extracted from the cladodes of *O. ficus-indica* belonging to the same plant but collected in different seasonal periods (winter and summer) and develop new extraction protocols that are able to improve the yield obtained and (b) evaluate the antioxidant potential of the fiber and study possible variations as a result of the extraction protocol chosen. The first objective was achieved (1) by measuring the amount of IF extracted from cladodes harvested in winter and summer (CW and CS, respectively) and (2) by modifying three variables involved in the fiber extraction protocol. To achieve the second objective, the following experiments were carried out: (1) measurement of the antioxidant potential of IF in CW and CS; (2) measurement of cellular reactive oxygen species; (3) measurement of the activity of some antioxidant enzymes; and (4) comparison of the polyphenol content in CW and CS. In conclusion, the results obtained showed that the IF extraction process can be improved, achieving a uniform yield regardless of seasonality; the antioxidant effect may vary depending on the extraction protocol.

## 1. Introduction

*Opuntia ficus-indica* (L.) Miller is a dicotyledonous plant in the category of Angiosperms, belonging to the *Cactaceae* family and to the genus *Opuntia,* which includes about 1500 species of cactus. The genus *Opuntia* is divided into four subgenera, *Cylindropuntia*, *Tephrocactus, Brasiliopuntia, and Platyopuntia*; the latter contains the group *Ficus*-*indica*, which includes *Opuntia ficus*-*indica* (L.) Miller [1]. *O. ficus-indica* is commonly called “prickly pear or nopal cactus” and is a plant that adapts well to arid and semi-arid climates, justifying it belonging to the tropical and subtropical plants [2]. For this reason, the geographical distribution of this plant includes Mexico, the country of origin, but also South and Central America, Africa, Australia, and countries in the Mediterranean area [3]. This succulent plant grows spontaneously, adapting to extreme climatic conditions, even in soils poor in minerals and low water availability [4]. The plant is arborescent, reaches a height between 3 and 5 m, and is characterized by an aerial part composed of morphological changes of the branches, called “cladodes”. The latter are green and covered with waxes, essential to reduce excessive perspiration of water, and generally obovate shape and variable dimensions (30–60 cm in length, 20–40 cm in width, and 19–28 mm thick) [5]. The cladodes are inserted one on top of the other, generating a characteristic tree shape without a real trunk and branches. The leaves present on the young cladodes are ephemeral, with a life cycle of only about 30 days. In place of the leaves, some very thin spines of a few millimeters that are yellow-brown color and very irritating appear and are called glochids. Glochids are a valid defense against many animals that would damage the plant [6]. A peculiarity of this genus is the stomata, which, contrary to what happens in most plants, remains close during the day; this significantly reduces water loss due to transpiration [7]. The flowers are hermaphrodites of intense yellow or yellow-orange color. The flowers form on the adult cladodes, and a fertile cladode can produce up to 20–30 flowers [8]. The fruit is an ovoid or pyriform berry with a fleshy and edible pulp. The skin (epicarp) assumes a variable color from yellow-orange to red, depending on the variety and the level of ripeness. The mesocarp and the endocarp consist of a very sweet and juicy pulp that envelops the numerous and small woody seeds. The root system is mostly superficial and develops in width rather than depth. The diffusion of *O. ficus-indica* is guaranteed by the birds, which disperse the seeds, eating their fruits [9]. The chemical composition of *O. ficus-indica* depends on many factors, including species, cultivar, environmental factors, climatic conditions, fertilization, maturity status, and crop management [10]. The compounds contained in *O. ficus-indica* [11] justify many of the beneficial activities carried out by this plant on human health, such as antioxidant, anti-inflammatory, anticancer, antimicrobial, and nutritional properties [12]. In this manuscript, we will consider only the cladodes of *O. ficus-indica* composed of water, polysaccharides, fiber, proteins, vitamins, fatty acids, sterols, mucilages, polyphenols, and minerals. The chemical composition of cladodes varies according to different factors, such as plant age, growing season, soil factors, and growth area [13]. In this respect, changes in protein content, pH, total sugars, dry matter, and conductivity have been observed depending on soil-specific profiles and climatic conditions [14,15].

Cladodes are responsible for retaining and absorbing water. For this reason, they are real “tanks” of water. Cladodes contain amino acids, such as isoleucine, threonine, asparagine, and alanine, ensuring 6.7–11.73% of protein content [16]. Another class of compounds contained in cladodes and responsible for beneficial activities are polyphenols, characterized by one or more hydroxyl groups connected to a benzene ring [17]. The identification of the polyphenols contained in the cladodes has been optimized through HPLC, and the most known compounds are quercetin, naringin, ferulic acid, piscidic acid, eucomic acid, quinic acid, malic acid, p-coumaric acid 3-O-glucoside, tricin, hydroxyl octadecadienoic acid, eicosanoic acid, rutin, kaempferol-rutinoside, and narcissin, among others [18,19]. Polyphenols in cladodes are responsible for antioxidant activity both in vitro and in vivo. Polyphenols, together with other components (saponins, sterols, lignans, and some vitamins [20]), play important protective roles in some metabolic diseases, such as metabolic syndrome, hypercholesterolemia, and obesity, as well as hypertension, asthma, and rheumatic pain [21]. 

Cladodes are also rich in mucilage, a complex organic substance that can absorb and retain the water, swelling enormously and ensuring the plants themselves the possibility of resisting desiccation and drought. Precisely for this reason, mucilage is also defined as “natural hydrocolloids” and is used as stabilizers, thickeners, and antioxidants in various food, pharmaceutical, and cosmetic applications [22]. Recent studies have defined mucilage as suitable in coatings used in the construction industry, as encapsulation agents, in the production of sustainable bioplastics, and in the manufacture of gluten-free foods [23].

Cladodes are excellent prebiotics and are able to modify and improve the composition of the gut microbiota [24]. Scientific research, in recent years, has highlighted the effective role of *O. ficus-indica* as a bioreactor to insert information and express compounds of interest. An example is provided in the paper by Akanni et al. [25], which highlights how the cultivation of yeasts *Candida utilis* and *Kluyveromyces marxianus* can improve the total protein content in an enzymatic cladode hydrolysate. Furthermore, a recent study [26] has shown the proliferation of the cladodes of *Hylocereus polyrhizus*, a plant belonging to the *Cactaceae* family, using a system of bioreactors, and these results could be applied to encourage the commercial micropropagation of this fruit crop.

Finally, in the cladodes, there is a high amount of fiber, which, retaining water, regulates its content, as well as the flow of calcium ions during prolonged drought [27]. Dietary fiber positively affects the physiology of both humans and animals. Soluble fibers, which dissolve in an aqueous solution, include pectins, mucilages, and certain types of hemicellulose and possess hypoglycemic, hypolipidemic, and hypocholesterolemic properties [28]. Insoluble fibers, which include cellulose and hemicellulose, play the role of maintaining the regulation of intestinal function, modulating peristalsis, controlling pH, and contributing to the prevention of gastrointestinal disease [29]. The fiber contained in the cladodes makes these organs highly functional, increasing their market value and making them competitive in the food industry. For these reasons, in addition to their rich composition, which identifies them as important promoters of nutritional properties, cladodes are also used in the medical, cosmetic, and pharmaceutical fields [30]. 

The main purposes of this scientific work are as follows: (a)To compare the IF extracted from the cladodes of *O. ficus-indica* belonging to the same plant but collected in different seasonal periods (IF from CW and CS: IF-CW and IF-CS, respectively); in this respect, new extraction protocols that are able to improve the yield of the fiber were developed.(b)To evaluate the antioxidant potential of the fiber and study possible variations as a result of the extraction protocol chosen.

A representation of the main organs of *O. ficus-indica* is shown in Figure 1.

## 2. Results

### 2.1. Method of Fiber Extraction from O. ficus-indica Cladodes

The methods used to extract fiber from *O. ficus-indica* are multiple. One method could be the enzymatic–gravimetric of McCleary [31], which involves the use of enzymes, such as hydrolases (thermostable alpha-amylase), followed by the addition of ethanol and acetone and filtration. Another widely used method provides pretreatment with NaOH or hydrogen peroxide [32]. We have chosen to extract the fiber without using chemical solvents; the three most harmless and environmentally friendly methods are as follows: (1)Extraction with water [33];(2)Extraction with ethanol [34];(3)Steam extraction in the presence of lemon juice [35].

In this study, extraction with water has been chosen, using the method of Cheikh Rouhou et al. [36] with some modifications.

### 2.2. Yield of Cladode Fiber Extraction

First, the yield of IF-CW and IF-CS were calculated: (1) IF-CW, February 2023; (2) IF-CS, August 2023.

The extraction yield calculation was as follows:Fiber yield (%) = (g of the dried extracted fiber/g of the dried cladode powder) × 100.(1) Yield IF-CW = (26 g/412 g) × 100 = 6.31% ± 1.3%.(2) Yeld IF-CS = (53 g/419 g) × 100 = 12.64% ± 2.1%.(1)

As can be seen from the results obtained, IF-CS was statistically higher than IF-CW. These results are shown in Figure 2. The fiber content, calculated in CW and CS, was similar to that described in the literature [37].

### 2.3. Improvement of the IF Extraction Protocol

The parameters involved in the main phases of IF extraction are (1) the temperature of the solvent; (2) the maceration time; and (3) the stirring speed during maceration. We decided to modify these variables one at a time and leave the other two unchanged to understand the real role of the variable considered. The original protocol provided the values described in Table 1.

The three modified extraction methods used have been described in Table 2.

The yield of the extracted IF was calculated with the same formula previously mentioned. The results obtained are shown in Figure 3. In panel a, the temperature variations showed no change in IF-CS yield. On the contrary, a higher temperature (120 °C) resulted in a significant increase in yield in IF-CW. In panel b, changes in IF yield, dependent on the maceration time, were studied. In particular, only IF-CS was sensitive to this variable, showing a statistically significant increase at a maceration time of 6 h. The variation of the stirring speed (rpm) during the maceration time did not lead to any change in the yield of the extracted fiber, neither in IF-CW nor in IF-CS. This result is shown in Figure 3, panel c. Finally, in panel d, the extracted fiber yields are shown after making the selected modifications to the extraction protocol (temperature of the solvent = 120 °C; maceration time = 6 h). The first graphic in panel d shows again the fiber extracted from CS and CW with the original protocol (it was already shown in Figure 2). In the second graph in panel d, the fiber was extracted after the use of a protocol in which the temperature of the solvent was maintained at 120 °C. The third graph in panel d shows a maceration time equal to 6 h. 

In light of the results obtained, this study was continued by choosing the extraction protocol modified for both IF-CW and IF-CS, as shown in Table 3 below.

### 2.4. Antioxidant In Vitro Activity of IF Extracted from O. ficus-indica

To measure the potential antioxidant properties of IF-CW and IF-CS, the ORAC test was used, and the results obtained are shown in Figure 4. As can be seen, a comparison between the original extraction method and the modified one has been made (panels a and b, respectively). In panel a, it is shown that IF-CW and IF-CS have antioxidant activity; the corresponding curves are placed between two curves with Trolox concentrations of 7.6 µg/mL and 15.25 µg/mL. However, the curves of IF-CW and IF-CS are closer to the Trolox curve of 7.6 µg/mL than 15.25 µg/mL. It is also interesting to note that IF-CS has a slightly higher antioxidant activity than IF-CW. The extraction of IF carried out with the modified protocol (panel b) has modified the trend of the curves, increasing the antioxidant activity of both CW and CS. Even if the respective curves are still posed between Trolox 7.6 µg/mL and 15.25 µg/mL, they are positioned significantly higher and close to 15.25 µg/mL, suggesting a stronger antioxidant potential. Once again, IF-CS possesses a higher antioxidant activity than IF-CW. 

Subsequently, we measured the antioxidant capacity of IF-CW and IF-CS on the Huvec cell line, a non-pathological endothelial line isolated from the vein of the umbilical cord, subject to oxidative damage induced by treatment with hydrogen peroxide. Experimentally, the cells were treated with hydrogen peroxide (100 µM, 20’) or pretreated with IF-CW or IF-CS (1 mg/mL, 24 h) and then exposed to hydrogen peroxide, as indicated. The concentrations and treatment times with IF-CW/IF-CS were the result of cell viability (Appendix A). Again, IF-CW and IF-CS were extracted with the original or modified protocol (panels a and b). The results are shown in Figure 5. The extraction method, using the original protocol, resulted in the formation of IF-CW and IF-CS, both of which can prevent the accumulation of ROS, generated by hydrogen peroxide. However, the antioxidant potential of IF-CS was significantly greater than IF-CW, as demonstrated in panel a. To the right of panel a, the respective quantification is shown. Panel b showed the antioxidant potential of IF-CW and IF-CS extracted with the modified protocol. The protection compared to the damage induced by hydrogen peroxide appeared more marked, and the results showed a significantly higher antioxidant activity than that evidenced by the same components extracted with the unchanged protocol. To the right of panel b, the respective quantification is shown.

### 2.5. The Activity of Primary Antioxidant Enzymes using Protocols Modified Compared to the Original Protocols

Mammalian cells express three basic categories of antioxidant enzymes, catalase (CAT), superoxide dismutase (SOD), and glutathione peroxidase (GSH-Px), which can ensure the survival of organisms when subjected to oxidative stress [33]. The parameter that can give us the biological impact of antioxidant enzymes is activity. Therefore, we measured the activity of CAT, SOD, and GSH-Px in Huvec cells. The cells were exposed to the IF of *O. ficus-indica* (1 mg/mL for 24 h), obtained using the original or modified protocols, and enzyme activity was measured several times (3, 6, 9, 12, 24, and 30 h). In particular, the activity of these enzymes was compared after exposure to the fiber, extracted both with the original and the modified protocols. In panels (a) and (b) in Figure 6, the activity of CAT is represented. As you can see, the fiber extracted with the modified protocols showed a statistically significant increase in the activity of the enzyme compared to the fiber extracted with the original protocols in both IF-CW and IF-CS. In panels (c) and (d), a similar result was found for the enzyme SOD and in panels (e) and (f) for the enzyme GSH-Px. We can conclude that modified IF extraction protocols increase the activity of primary antioxidant enzymes.

### 2.6. DPPH Free Radical Scavenging Activity

The slightly modified 1,1-diphenyl-2-picrylhydrazyl (DPPH) assay was used to assess the scavenging activity of IF coming from *O. ficus-indica*. The results obtained were expressed as inhibition % and IC_50_ value, representing the concentration of the extract necessary to scavenge the 50% DPPH radicals. The inhibitory concentration IC_50_ of IF-CS and IF-CW was calculated after measuring the absorbance. IF-CS has been shown to have a strong antioxidant capacity that is more effective than IF-CW. IC_50_ of IF-CS was 2.286 ± 0.064 mg/mL, while IF-CW was 4.426 ± 0.032 mg/mL (Figure 7, panels a and b). This figure shows only the results obtained by IF extracted with the modified protocol.

### 2.7. Total Polyphenol Content

Next, we wanted to quantify the polyphenols contained in IF obtained with both protocols. In particular, from the different gallic acid concentrations, we had the respective quantification and, as can be seen in Figure 7, panel c, there was no significant difference in polyphenol content between CW and CS (with both extractive protocols used).

### 2.8. Effects of IF-CW and IF-CS on Cell Proliferation

To evaluate the effects of IF-CW and IF-CS on Huvec cells, experiments on cell proliferation have been carried out (Appendix A). In panel a, the cells were treated with different concentrations of IF-CW and IF-CS as indicated, and the proliferation was evaluated. As can be seen, concentrations 0.5, 1, 2, 5, and 10 mg/mL of IF-CW and IF-CS had no impact on cell proliferation. A significant decrease in viability was instead appreciated, increasing gradually, from the 15 mg/mL concentration onwards. These results have highlighted that IF-CW is more harmful than IF-CS. In panel b, the ability of IF-CW and IF-CS to prevent damage induced by treatment with lipopolysaccharide (LPS), a toxic component of the external cell membrane of the gram-negative bacteria, is shown. The cells were pretreated with IF-CW and IF-CS (0.5, 1, 2, 5, and 10 mg/mL) for 24 h and then treated with LPS 1 µg/mL for 24 h. As can be seen, IF-CW and IF-CS at a concentration of 1 mg/mL can significantly prevent LPS-induced damage. In these results, represented in Figure 1, IF-CW and IF-CS were extracted using the original protocol. However, no significant change was appreciated when the compounds of interest were extracted with the modified protocol.

## 3. Materials and Methods

### 3.1. Plant Materials and Sample Preparation

The cladodes of *Opuntia ficus indica* were collected in Roccelletta di Borgia, Calabria, Italy, latitude 38°87′48″ N and longitude 16, 58°30′53″ E, from the same plant in two different seasonal periods: (1) cladode winter (CW, February 2023, temperature 8.9 °C) and (2) cladode summer (CS, August 2023, temperature 36.4 °C). The collected cladodes were mature (28–34 cm long, 18–21 cm wide, and 2 cm thick). The cladodes were washed with distilled water, the plugs were removed, and they were cut into pieces of about 1 cm. Subsequently, the cladodes were dried in a laboratory oven (ENCO, Venice, Italy) at a temperature of 40 °C for 4 days. The dried pieces were pulverized with a grinder (Retsch SM 2000, Burladingen, Germany) and stored in suitable hermetically sealed bags, at −20 °C until use.

### 3.2. Fiber Extraction Protocol

Fiber extraction from *Ficus Indica* cladode powder was carried out by choosing water as the sole extraction solvent using the method of Cheikh Rouhou et al. [36] with some modifications. The chosen solid/liquid ratio was = 1/30 (*w*/*w*). Cladode powder was mixed with solvent (water) at a temperature of 60 °C for 3 h (maceration time) and under constant stirring (120 rpm). At the end of the maceration time, the solution was centrifuged (6500× *g* for 10 min) five times, each time resuspending the solid phase in 100 mL of water at 40 °C). Finally, the resulting solid phase was dried in the oven at 40 °C and stored at 4 °C. 

### 3.3. Extraction Yield

The yield of the obtained fiber was expressed in % and was calculated by dividing the weight (g) of the dried extracted fiber by the weight (g) of the dried cladode powders [32] as follows:Fiber yield (%) = (Ef/Cp) × 100(2)

### 3.4. Oxygen Radical Absorption Capacity (ORAC) Assay

The antioxidant activity of IF-CW and IF-CS was determined by the Oxygen Radical Absorbance Capacity (ORAC) test. This method measures the antioxidant activity of a sample by evaluating the transfer of a hydrogen atom. In particular, fluorescein fluorescence loss (used as a probe) is measured over time. This fluorescence is due to the formation of peroxylic radicals, following the spontaneous degradation of 2,2′-azobis-2-methyl-propanimidamide and dihydrochloride (AAPH), which occurs at 37 °C. The peroxylic radical oxidizes the fluorescein, causing a gradual loss of the fluorescence signal. Antioxidants suppress this reaction and inhibit the loss of the signal. 6-Hydroxy-2,5,7,8-tetramTethylchroman-2-carboxylic acid (Trolox) is a water-soluble analog of vitamin E that inhibits fluorescence decay in a dose-dependent manner. Fluorescein and AAPH solutions were prepared in PBS (pH = 7.0) at concentrations of 0.02 mg/mL and 59.8 mg/mL, respectively. In contrast, Trolox was made in PBS (pH = 7.0) at concentrations of 7.65, 15.25, 30.5, and 61 μg/mL. Finally, IF-CW and IF-CS were used at a concentration of 1 mg/mL. The evaluation of the fluorescent decay for fluorescein was conducted using a microplate reader, where excitation and emission wavelengths of 485 and 520 nm were used, respectively, at 37 °C. Measurements were made in triplicate every 2 min for 1.5 h, and the data obtained from fluorescence vs. time curves are reported as the average antioxidant efficacy of the antioxidant compound. A regression equation was constructed by comparing the net area under the fluorescein decay curve and the Trolox concentration. The area below the curve has been calculated with the following equation:
i = 90        AUC = 1 + Σ f1/f0  i = 1(3)

### 3.5. Cell Cultures

The human endothelial cell line (HUVEC) was isolated from the vein of the umbilical cord and acquired from the American Type Culture Collection (Sesto San Giovanni, Milan, Italy) and kept in F-12 K Medium, suitably completed with endothelial cell growth supplement and fetal bovine serum. The cell line was cultivated in a 5% humidified CO_2_ atmosphere and maintained at 37 °C. The medium was changed every 2–3 days, and when the cell lines reached 60% confluence, they were treated with IF-CW and IF-CS (original and modified protocol) 1 mg/mL for 24 h. 

### 3.6. Measurement of ROS in Cells

H_2_ DCF-DA is a molecule that easily enters cells and is subsequently cleaved by intracellular esterase to form H_2_ DCF, which is deprived of an acetate group and is no longer able to leave the cells and, if oxidized, binds to the ROS, forming the compound highly fluorescent DCF. The quantification of the DCF probe provides the content of ROS in the cell. Huvec cells were seeded in 96-well microplates with a density of 6 × 10^4^. The following day, they were pretreated with IF-CW or IF-CS at the concentration of 1 mg/mL for 24 h. At the end of the treatment, the growth medium was replaced by a phenol-free fresh medium containing H_2_ DCF-DA (25 µM) and, after 30 min of exposure to 37 °C, the cells were washed with PBS (to remove excess H_2_ DCF-DA), centrifuged, resuspended in PBS, and exposed or not at H_2_ O_2_ (100 µM, 20 min). The fluorescence was evaluated by cytometric analysis (FACS Accury, Becton Dickinson, Franklin Lakes, NJ, USA).

### 3.7. Measurement of Cell Proliferation through the MTT Test

The colorimetric assay based on the use of bromide of 3-(4,5-dimethyl-2-yl)-2,5-diphenyltetrazole (MTT) was used to assess cell proliferation. This test is based on the indication that live cells possess mitochondria with active enzymes, which can reduce MTT, resulting in colorimetric variation. Consequently, the measurement of MTT reduction provides information on cell viability and metabolic activity. Experimentally, 8 × 10^3^ cells were plated in 96-well plates and, after 24 h, the culture medium was replaced with fresh medium containing IF-CW or IF-CS at several concentrations (Figure 1). After 24 h, the medium was replaced with a phenol-free medium containing a solution of MTT (0.5 mg/mL). After 4 h of incubation, 100 µL of 10% SDS was added to each well to solubilize the formazan crystals. Finally, the optical density was measured at wavelengths of 540 and 690 nm using a spectrophotometer reader (xMARK^TM^ Microplate Absorbance Spectrophotometer, Bio-Rad, Segrate (MI), Italy).

### 3.8. Catalase, Superoxide Dismutase, and Glutathione Peroxidase Activities

Following the treatments, carried out as described, the Huvec cell line was collected to measure the activities of CAT, SOD, and GSH-Px. For this purpose, the respective kits were used (UK, Cambridge, and Abcam), according to the manufacturer’s instructions. CAT can convert hydrogen peroxide to oxygen and water. In this assay, unconverted H_2_ O_2_ reacts with the Oxired probe to produce a product that can be measured at 570 nm, whose activity is inversely proportional to the signal. There are three isoforms of SOD, all highly compartmentalized, although the most important is in the cellular cytoplasm and accounts for about 90% of isoforms. The function of SOD is able to convert superoxide radicals into hydrogen peroxide and molecular oxygen (O_2_). In the kit used to measure SOD activity, superoxide anions react with a specific probe to produce water-soluble formazan dye, which is detected by measuring absorbance at 450 nm. The higher the SOD activity in the sample, the lower the formazan dye produced. Finally, GSH-Px converts hydrogen peroxide into water in a reaction involving glutathione and determining NADPH consumption. The reduction in NADPH is measured at 340 nanometers and is proportional to Gpx activity.

### 3.9. Antioxidant Activity through the DPPH Assay

The antioxidant activity of the water-dissolved extract from the cladodes of O. *ficus-indica* was measured using the stable radical 2,20 -diphenyl-1 picrylhydrazyl (DPPH) at a concentration of 4 mg/100 mL [34]. The reduction in absorbance, visible as a change of color from purple to yellow, was measured. Experimentally, 850 µL of the DPPH solution was added to 50 µL of various extract concentrations (0, 1, 2, 3, 4, and 5 mg/mL), keeping the mixture in the dark for 20 min. Subsequently, the absorbance was read by a UV-Vis spectrophotometer (Multiskan GO, Thermo Scientific, Denver, CO, USA) at 517 nm at room temperature. The results obtained were expressed as % inhibition value and IC_50_. The latter represents the concentration of powdered matter needed to remove 50% of DPPH radicals.

### 3.10. Measurement of Total Polyphenols through the Folin–Ciocalteu Assay

The measurement of total polyphenols, contained in the powder of *O. ficus-indica* cladodes, was carried out using the modified Folin–Ciocalteu colorimetric test [33]. Gallic acid solutions (0, 25, 50, 100, 200, and 300 µg/mL) were used as the reference standard for drawing a calibration curve. Experimentally, 1 g of cladodes powder from *O. ficus-indica* was mixed with 20 mL ethanol/water 50:50 (*w*/*w*). The mixture obtained was stirred for 24 h at 20 °C. For the measurement, 400 μL of the mixture was put into a cuvette and 0.8 mL of Folin–Ciocalteu reagent diluted 10 times was added, having taken care to shake thoroughly. After 3 min, 0.8 mL of sodium carbonate 7% (*w*/*v*) was added, and the mixture obtained was allowed to stand for 2 h with intermittent stirring until color developed. The relative absorbance was measured at 760 nm with a Prisma V-1200 Spectrophotometer and the phenolic content was determined from the linear equation of the standard curve prepared with different concentrations of gallic acid. The total phenolic content of the extract was expressed in mg gallic acid equivalent (GAE)/g dry weight.

## 4. Discussion

Food safety requires an adequate intake of nutrients in the daily diet. Nowadays, the global situation is sharply divided into two categories: on the one hand, food shortage leads to undernutrition and insufficient habitual consumption; on the other, the extreme availability of food induces excessive consumption and obesity. This concept can be summed up in the observation that the world is divided between undernutrition and hypernutrition/obesity [38,39]. For this reason, it has become increasingly important to eat foods with high nutritional potential, easily available, and affordable [40]. In recent decades, there has been a growing belief that foods with these characteristics can be represented by products of plant origin. Plants generate easily available foods, frequently with low economic impact and a rich nutrient composition [41]. Moreover, many phytocomplexes are called “functional foods” because, in addition to possessing specific nutritional properties, they have demonstrated the ability to positively influence one or more physiological functions, improving health [42]. Numerous studies in the literature have highlighted the chemical composition of *O. ficus-indica*, indicating that this plant contains high values of important nutrients. Cladodes are rich in vitamins (vitamin C, α-tocopherol, β-carotene, indicaxanthin, and betanin) [43], amino acids (alanine, arginine, asparagine, aspartic acid, glutamic acid, glutamine, glycine, serine, threonine, tyrosine, tryptophane, and valine) [44], sterols (campesterol, stigmasterol, β-sitosterol) [45], and minerals (Ca, Na, K, Mg, Fe, Cu, Zn, Mn, and Ni) [46]. In general, fiber has long been considered an important element in glycemic control and fat absorption due to its gastric emptying properties [47,48,49]. The chemical composition of *O. ficus-indica* includes fair amounts of fiber, capable of improving glucose absorption [50,51], regulating carbohydrate metabolism and digestion, preventing hypercholesterolemia, and modulating cholesterol activity [52]. For these reasons, cladodes are known to provide numerous beneficial properties for human health [53]. 

First of all, we have quantified the IF extracted from the cladodes of the same plant of *O. ficus-indica* collected in two different seasonal periods: winter and summer. The comparison of the yield of IF from CW and CS showed significantly higher content in CS. The observed variability could be explained by the different seasons in which cladodes were collected [54,55]. In particular, it is important to remember that the collection of CW occurred in winter with a temperature of 9 °C, while CS was taken in summer with a temperature of 36 °C. A temperature difference of 27 °C between CW and CS collection could explain the different IF concentrations present in the two samples. Indeed, although CW and CS were collected from the same plant, the external temperature could justify the observed differences in composition [56]. Next, we have modified the IF extraction protocol to improve the yield of the fiber. The method chosen and used for fiber extraction was that of Cheikh Rouhou et al. with some modifications. This choice was justified by the desire to use a protocol without the use of chemical solvents, which could cause both major changes in fiber composition and a negative impact on the environment, contributing to increased pollution [57,58,59]. The modified variables in the new protocols have been well described in Table 1, Table 2 and Table 3. 

The results obtained suggested that higher temperature favors IF extraction from CW. In particular, a temperature of 120 °C is needed, instead of 60 °C, to increase the yield of fiber extracted from CW from 6.3% to 16%. This could be explained by the different rates of plant metabolism, which slow down in winter and accelerate in summer [60]. Therefore, such a high temperature could cause the acceleration of metabolism and increased fiber yield. It should not be forgotten that this plant grows easily and spontaneously in very arid or torrid areas, demonstrating an excellent ability to adapt to excessively high temperatures. A higher temperature could presumably promote and enrich the content of many components, including fiber [61]. However, it remains to be studied whether 120 °C can already be considered a temperature harmful to the vital mechanisms of the plant. 

Conversely, an increase in the maceration time during the extraction of the fiber has determined a significant increase in the yield of the fiber contained in CS. Maceration is an extraction process in which a solid component is kept immersed in a liquid solvent for a variable time, depending on the type of solvent used and the compounds to be extracted. Experimentally, the vegetable ingredient of interest must, first of all, be chopped into small pieces to increase the contact with the solvent and allow it to penetrate the innermost cells. At the same time, the sample should not be pulverized in order not to lose its volatile active ingredients and to make filtration possible at the end of the process. Once the ideal size is reached, the solid component is immersed in the solvent for a variable time. At the end of the maceration time, the macerate is gently filtered and used, while the insoluble residue is eliminated [62].

The maceration time can vary, as already specified, from a few hours to several weeks. Although maceration usually takes place at room temperature, it has recently been shown that temperature and maceration time can greatly affect the extractive yield [63,64]. 

The fiber content in CS was collected in the summer with an external temperature of 36.4 °C. If to this condition we add a longer maceration time, we obtain a fiber yield ranging from 12.64% to 19%. All subsequent experiments were conducted with both protocols to appreciate the differences.

The antioxidant potential of the fiber has been tested, and the pretreatment with IF-CW or IF-CS significantly reduced the accumulation of ROS. In all experiments conducted and displayed, IF-CS showed a stronger antioxidant power than IF-CW, both with the original extractive protocol and with the modified one. Again, these differences could be the consequence of the different seasons in which cladodes were collected [65,66,67]. 

The antioxidant effect could be generated by the presence of polyphenols, which can act as electron donors, converting free radicals into more stable products and showing a scavenging effect [68]. However, unlike the IF content between CW and CS, there are not the same variations in the polyphenol content, which are present in both fractions in the same quantities. In addition, the different extraction protocols described did not affect the amount of polyphenols, as shown in Figure 7, panel c. 

Since the modified fiber extraction protocol seems to increase the antioxidant power compared to that obtained with the original protocol, the hypothesis formulated is that the parameters of the modified protocol are responsible for antioxidant improvement. Therefore, the continuation of this experimental work requires further experiments to confirm and explain this hypothesis.

## 5. Conclusions

The results obtained in this paper lead to three important conclusions: (1)Cladodes are a source of fiber;(2)There are characteristics of the fiber extraction protocol that, if modified, may increase the fiber yield;(3)IF obtained from modified extraction protocols increases the antioxidant potential and reduces cellular oxidative damage.

## Figures and Tables

**Figure 1 plants-13-00544-f001:**
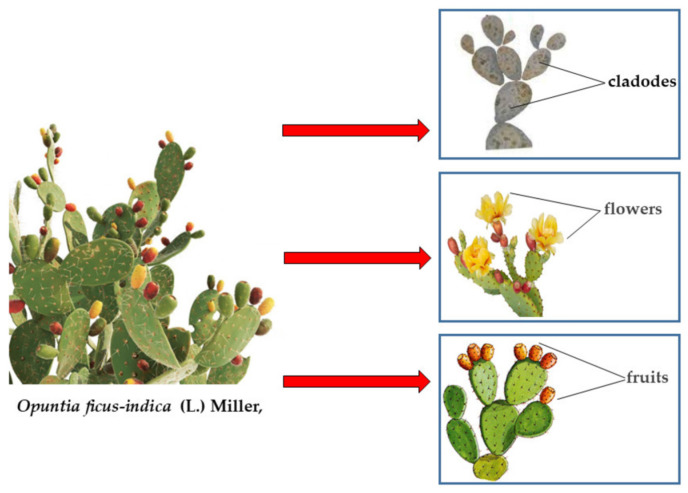
Representation of the main organs of *O. ficus-indica*.

**Figure 2 plants-13-00544-f002:**
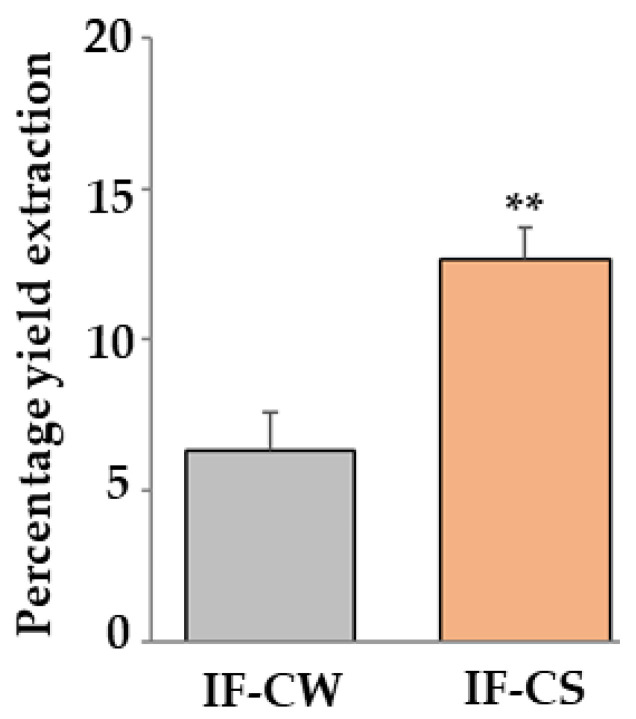
Comparison between CW and CS. As can be seen in Figure 2, a significant variation in IF yield between CW and CS was appreciated. In particular, IF-CS was higher than IF-CW. Three independent experiments were carried out, and the values are expressed as the mean ± standard deviation (sd). ** denotes *p* < 0.01 vs. CW. Student’s *t*-test was applied.

**Figure 3 plants-13-00544-f003:**
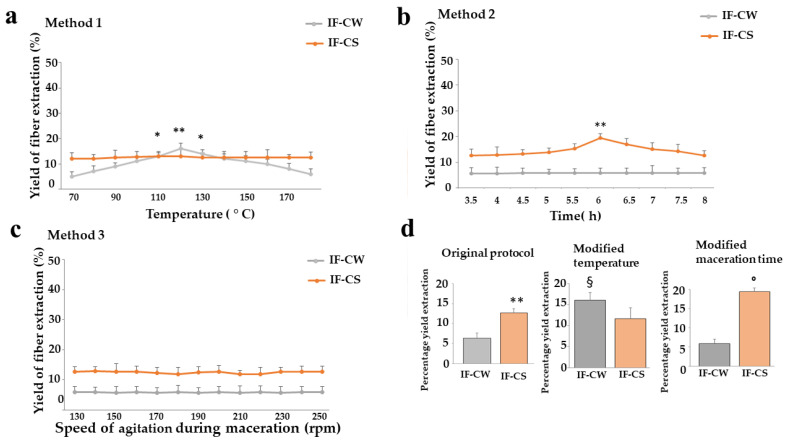
Modifications that improve IF extraction yield. The described modified extraction methods 1, 2, and 3 are shown in panels (**a**–**c**), respectively. Panel (**d**) shows the IF yield variations obtained in both IF-CW and IF-CS using modified protocols. Three independent experiments were carried out, and the values are expressed as the mean ± standard deviation (sd). In panels (**a**,**b**), * denotes *p* < 0.05 vs. the basal condition; ** denotes *p* < 0.01 vs. the basal condition. In panel (**d**), ** denotes *p* < 0.01 vs. IF-CW; § denotes *p* < 0.05 vs. IF-CW obtained with the original protocol; **°** denotes *p* < 0.05 vs. IF-CS obtained with the original protocol. Analysis of variance (ANOVA) was followed by a Tukey–Kramer comparison test.

**Figure 4 plants-13-00544-f004:**
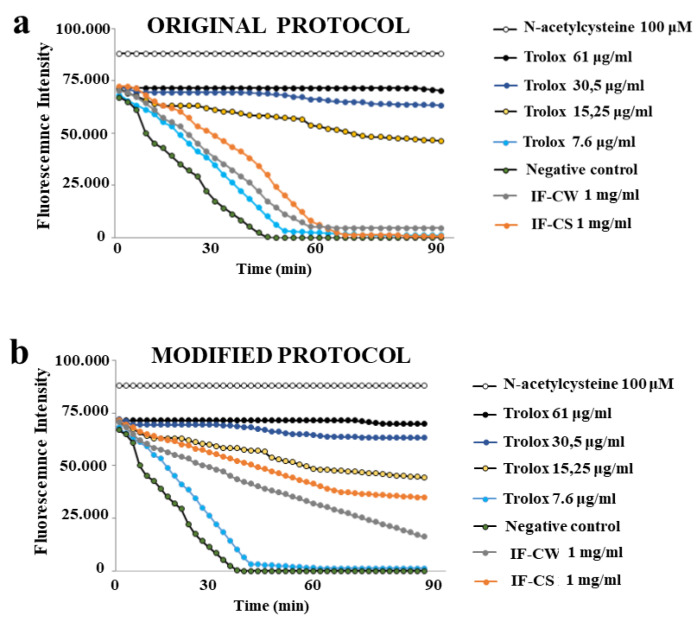
ORAC assay decay curves for Trolox, IF-CW, and IF-CS. The Trolox concentrations generated a linear relationship with the net area under the curve (AUC). Subsequently, samples IF-CW and IF-CS (1 mg/mL) were analyzed. In this test, the maintenance of the fluorescence signal is indicated as the area under the curve (AUC) measured over time (0–90 min). Relative values of IF-CW and IF-CS were obtained by comparing their AUC with that of antioxidant curves. In panel (**a**), the curves of IF-CW are represented, and IF-CS is extracted with the original protocol. On the contrary, panel (**b**) shows the curves of the same samples extracted with the modified protocol.

**Figure 5 plants-13-00544-f005:**
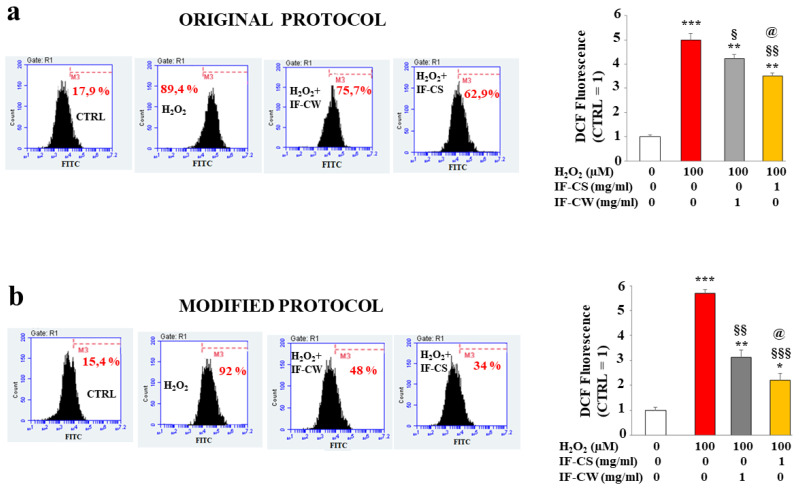
Antioxidant effect of IF-CW and IF-CS in Huvec cells. In panel a, the antioxidant activity exerted by IF-CW and IF-CS extracted from *O. ficus-indica* with the original protocol has been measured. The measurement of reactive oxygen species (ROS) is shown through cytofluorometer analysis. In particular, the cells were treated with hydrogen peroxide (100 µM for 20’), and this treatment highlighted the production and accumulation of ROS, as evidenced by the shift on the right of the fluorescent cellular peak, compared to untreated cells. Pretreatment with IF-CS and IF-CW (1 mg/mL, for 24 h) significantly reduced the ROS generated by hydrogen peroxide. In panel (**a**), on the left, every single box is generated following the reading of the fluorescence of the cells; in particular, the *x*-axis represents the fluorescence of fluorochrome FITC linked to our fluorescent probe, while the *y*-axis is relative to the number of cells that we decided to acquire (10,000). At the top of each box, there is a marker (M3), which is arbitrarily drawn in the control and kept the same for all other samples. The part of the peak included in M3 is indicated by a numerical percentage. On the right of panel a, the respective quantification, obtained from the comparison of the percentages, is represented. The control percentage is arbitrarily made equal to 1, and the other values are related to it. In panel (**b**), the same compounds, but extracted with the modified protocol, were evaluated under experimental conditions similar to those described in panel (**a**). Three independent experiments were performed, and the values were expressed as the mean ± sd. * denotes *p* < 0.05 vs. the control; ** denotes *p* < 0.01 vs. the control; *** denotes *p* < 0.001 vs. the control. § denotes *p* < 0.05 vs. H_2_ O_2_; §§ denotes *p* < 0.01 vs. H_2_ O_2_; §§§ denotes *p* < 0.001 vs. H_2_ O_2_; @ denotes *p* < 0.05 vs. IF-CW. Variance analysis (ANOVA) was followed by a Tukey–Kramer comparison test.

**Figure 6 plants-13-00544-f006:**
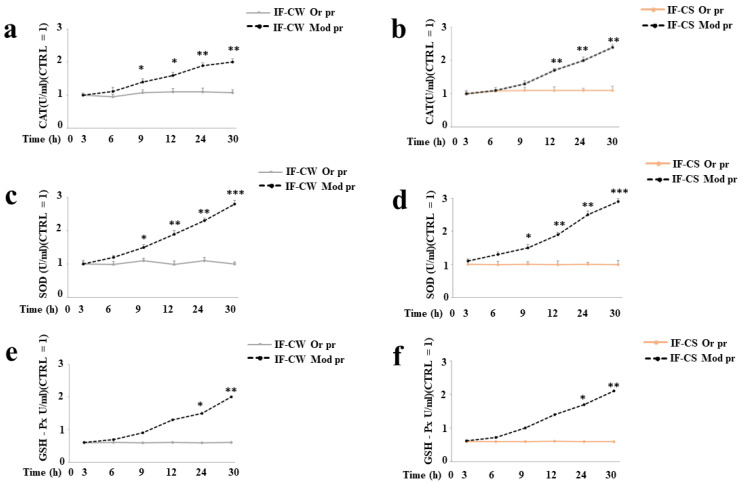
The activity of CAT, SOD, and GSH-Px following exposure with IF-CW and IF-CS (1 mg/mL). In panels (**a**,**b**) in Figure 6, the activity of catalase is represented. The extraction of the fiber was carried out both with original (Or Pr) and modified protocols (Mod pr). Enzyme activity was followed at different times (3, 6, 9, 12, 24, and 30 h). Figure 6 also reports the activity of SOD and GSH-Px under the same experimental condition panels (**c**–**f**). Three independent experiments were carried out, and the values are expressed as the mean ± standard deviation (sd). * denotes *p* < 0.05 vs. IF obtained using original extraction protocols; ** denotes *p* < 0.01 vs. IF obtained using original extraction protocols; *** denotes *p* < 0.001 vs. IF obtained using original extraction protocols. Analysis of variance (ANOVA) was followed by a Tukey–Kramer comparison test.

**Figure 7 plants-13-00544-f007:**
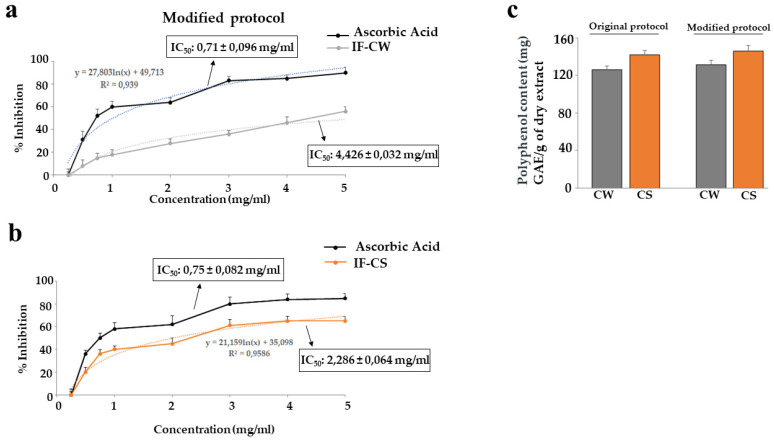
The inhibitory concentration IC_50_ of IF-CS and IF-CW. In panels (**a**,**b**), the antioxidant capacity of IF-CW and IF-CS is shown. As you can appreciate, IF-CS has a higher antioxidant capacity than IF-CW. In panel (**c**), the polyphenol content in both fractions obtained by original and modified protocols is highlighted. Three independent experiments were carried out, and the values are expressed as the mean ± standard deviation (sd).

**Table 1 plants-13-00544-t001:** Description of the original fiber extraction protocol from *O. ficus-indica*.

ORIGINAL PROTOCOL	Temperature of the Solvent	Time of Maceration	Stirring Speed during Maceration
	**60 °C**	**3 h**	**120 rpm**

**Table 2 plants-13-00544-t002:** Description of three potential and innovative methods (method 1, method 2, method 3) used for IF extraction. Each of the methods considered changes in only one parameter of the fiber extraction method and leaves the others unchanged.

	Temperature of the Solvent	Time of Maceration	Stirring Speed during Maceration
**Method 1:**	**70–180 °C**	3 h	120 rpm
**Method 2:**	60 °C	**4–8 h**	120 rpm
**Method 3:**	60 °C	3 h	**130–250 rpm**

**Table 3 plants-13-00544-t003:** Selected modified fiber extraction protocol from *O. ficus-indica*.

MODIFIED PROTOCOL	Temperature of the Solvent	Time of Maceration	Stirring Speed during Maceration
	**120 °C**	**6 h**	**120 rpm**

## Data Availability

Data are contained within the article and Appendix A.

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
