# Peer review of "Antioxidant Activity and Seasonal Variations in the Composition of Insoluble Fiber from the Cladodes of Opuntia ficus-indica (L.) Miller: Development of New Extraction Procedures to Improve Fiber Yield"

_plants, 2024, doi:10.3390/plants13040544_

Round 1
Reviewer 1 Report (Previous Reviewer 2)
Comments and Suggestions for Authors
In general changes have improved the content of this interesting report. However, Conclusions are still rather poor; in spite of that, this report is acceptable for publication.
Comments on the Quality of English LanguageAcceptable
Author Response
Reviewer 1
In general changes have improved the content of this interesting report. However, Conclusions are still rather poor; in spite of that, this report is acceptable for publication.
Thank you for your valuable advice.
The revision of English was done as well.
Reviewer 2 Report (New Reviewer)
Comments and Suggestions for Authors
The present article discusses the antioxidant activity of the insoluble fibers fraction derived from Opuntia ficus-indica (L.), evaluated within two seasonal periods. It also presents a new protocol with which a higher amount of fibers is obtained.
Authors performed various experiments to prove differences between the collected samples and the extraction procedure chosen.
Regarding the manuscript I have the following comments:
The English language used is often confusing. This makes very difficult the understanding of the intended meaning. Authors should carefully read the manuscript, correct grammatical mistakes and improve sentences clarity throughout the text. Some examples (but not limited only to these) are given below:
Lines 55-56: “having a duration of about 30 days”. You mean their life cycle?
Line 60: “open at night and remain closed by day”. Please correct to…remain close during the day.
Line 93: “metabolic pathologies” . Please rephrase to metabolic diseases or disorders.
Lines 94-100: This is a very abrupt beginning. Please add an introductory sentence
Lines 103-105: please correct the sentence
Lines 152-153: “was like the percentage described in the literature”. Please correct to …was similar to that described in the literature
Lines 170-171: the sentence does not make sense. Please rephrase.
Lines 278-279: “obtained using different protocols, as indicated” . As indicated by who?
Line 324: “damage to cell proliferation”. I think a more suitable expression is the following: “had no impact on cell proliferation”, or something similar.
Line 445: What do you mean by dust?
Other comments
Line 73-74: What is the meaning of this sentence? I mean why did you write it? Please delete, rephrase or explain better.
Lines 170-178: Information presented herein does not add any new information compared to that of table 2. I think this paragraph is not necessary.
Lines 248-249: please correct the legend of figure 5
Lines 250-270: Is this a part of the main text?
Line 334: Is this figure part of the supplementary material?
Lines 393-394: “At the end of the treatment time, appropriate tests have been carried out.” What do you mean?
Line 437: Please replace the word powder with the word extract and also indicate the solvent used to dissolve the powdered material.
Lines 486-495: In these lines you refer to the different yield?
Lines 533-537: This paragraph is a repetition of tables 1 and 2. I suggest to delete
Lines 473-474: “The study of the chemical composition indicated that O. ficus-indica contains high values of important nutrients”. Which study? Studies of other researchers?
Lines 570-589: The hypothesis stated by the authors is not clear. You should explain how (if) the change of temperature and extraction time influence bioavailability of the compounds. Since you do not know the chemical profile of the extracts, I think such a hypothesis is very risky. My opinion is to focus on the different values between the extracts, obtained by the antioxidant activity assays.
Lines 597-598: since not a method of the chemical identification of the compounds presented in the extracts was performed you cannot say that the plant contains several compounds.
General comments
The discussion is mostly a repetition of the results. I think you should strengthen this section by adding more information regarding comparison of your results respect to other studies.
Lastly, please explain this original protocol. What do you mean by original? Is it a protocol developed by who? Is it a protocol developed by your team and then modified? Which solvent was used for the extraction process at the original protocol? It is very confusing this “original protocol”
Comments on the Quality of English LanguageExtensive editing of English language is required
Author Response
Please see the attachment.

Reviewer 3 Report (New Reviewer)
Comments and Suggestions for Authors
The authors studied the antioxidant potential of Opuntia ficus indica’s insoluble fiber extract prepared from the summer and winter collected cladodes by various extraction methods. The fibers of the cladode extracts are also significant as functional food compounds, so this study can be considered important and interesting for the readers of the journal Plants. However, some questions need to be answered and some corrections/improvements need to be made as follows (points 1-9):
1) Lines 75, 77: “…main metabolites are water, polysaccharides, fiber, proteins, vitamins, fatty acids, sterols, mucilages, polyphenols, and minerals …” Water and minerals are not plant metabolites, so this sentence needs to be corrected.
2) Line 81: “…Cladodes are formed by molecular networks…” The meaning of this sentence is not clear, it needs corrections.
3) Fig. 2 should be deleted: all data of the figure are detailed in the text.
4) Lines 161-162: “…The main steps of IF extraction are three: (1) the temperature of the solvent; 2) the time of maceration; 3) the stirring speed during maceration…” These (temperature, time, stirring speed) are not “steps”: This sentence needs to be corrected.
5) Fig. 4. should be simplified: The curves of N-acetylcysteine, Trolox (in various concentrations) and neg. control seems to be the same in panels A and B, their curves should be removed from panel B (unnecessary repetition).
6) Lines 235-237: “…The concentrations and treatment times, with IF-CW/IF-CS and hydrogen peroxide, were the result of cell viability (Supplemental Figure 1) and previous experiments performed, respectively.…” This sentence is difficult to understand, it needs to be improved.
7) Figure 5: The caption of the Fig. 5 needs to be completed (to make the picture understandable).
8) Figure 6: Positive control is missing. Why wasn't a positive control used in this experiment?
9) Section 2.9: Effects of IF-CW and IF-CS on cell proliferation. Extracts prepared with original protocol and modified protocol were used in all previous tests. Why weren't both extracts used in this experiment as well?
In conclusion, the manuscript deserves publication after a major revision detailed above.
Round 2
Reviewer 2 Report (New Reviewer)
Comments and Suggestions for Authors
The authors have satisfactorily addressed my concerns in the revised version of the manuscript.
I have no further suggestions.
Reviewer 3 Report (New Reviewer)
Comments and Suggestions for Authors
The authors improved their manuscript. It can be accepted for publication.
This manuscript is a resubmission of an earlier submission. The following is a list of the peer review reports and author responses from that submission.
Round 1
Reviewer 1 Report
Comments and Suggestions for Authors
The paper "Seasonal variations in fiber composition from Opuntia ficusindica (L.) Miller cladodes: development of new extraction procedures to improve fiber yield" is a fundamental investigation into the fibre yield, antioxidant activity and polyphenolic composition of Opuntia ficusindica (L.) cladodes harvested in winter and summer. However, the article does not make a substantial contribution to the field, as there are already numerous existing articles characterising Opuntia ficusindica (L.) cladodes. The lack of a clear hypothesis and of significant challenges in characterisation further reduce the impact of the article. The current separation of the results into a separate section is problematic and the poor organisation of the article makes it difficult for the reader to navigate. To improve clarity, the article should be restructured by combining the results and discussion into a single section. In summary, the article is not recommended for publication due to its lack of originality, inadequate quality, limited significance and insufficient scientific rigour.
Author Response
Dear Reviwer,
thanks for your valuable advice. As you can see the paper has been remodeled in a clearer form and numerous results have been added.
I have better listed the main objectives and hope that in this form it can be considered ready for publication.
Reviewer 2 Report
Comments and Suggestions for Authors
This is an interesting type of research. With climatic changes which have already arrived; with water limitations in most agricultural areas of the world; and with the increasing necessity of the populations to improve the daily diet with food components and ingredients with outstanding nutrition, nutraceutical and even medical message; then, cacti plants and especially nopal and overall Opuntias genus have become the crop for the 21st century, as it has been classified by international organization.
Therefore, the present report may be very useful to enrich potential options. However, there are important changes before its possible acceptation for publication in Plants.
Objectives - They are acceptable. However, there other equally important objectives which are not covered here; and that they deserve, at least, not to be ignored; on the contrary, authors are suggested to consider. There is one study published in a highly cited journal about changes of the plant in terms of composition, and especially over cladodes, during ripening. Additionally, as important as selecting two different stations of the year, it is equally important, and I do believe that much more important as objective, to study the influence of the geographical region on the composition, especially again, of cladodes; it should be considered here the same cultivar.
Abstract, results and discussion, and conclusions - Authors need to correct the use of cladodes as biological wastes. Cladodes are not wastes at all. Nopal cultivars are not properly used based in their high potential in view of their genetic and agronomical characteristics; but they are not exactly wastes; and it is the same for cladodes. At the end, they are materials not fully used according to the cited potentialities.
For the information to authors, cladodes are used in Mexico and Central America, and also in the Mexican - Hispanic populations in the US as very frequent main dishes, and ingredients. And they are called "nopalitos". There is two or three publications from our group, which I should not cite for ethical reasons, about agronomic, reproduction in the field, and composition based on age of the plant ( and consequently age of cladoes) and their influence on such "nopalitos" (cladodes). Therefore, avoid the use of "wastes" or re-difine terms because under such characteristics most cultivar components are "wastes" once they are not fully used.
Cladodes - Authors are suggested to mention that cladodes contain mucilages and liquid components which, according to the cultivar, etc, they have outstanding nutraceutical and medical components which may be extracted and potentially commercialized.
Nopal plant as a bioreactor - There is one or two published studies, based on the performance of the plant under non-adequate environments, which may be genetically modified to insert into them, especailly in the cladodes, the information to express or over-express compounds of pharmaceutical importance; it means, to use the plant as a bioreactor. Authors may use this publication in the Introduction section, as a background.
Conclusions - They are really very poor. Authors need to underline, based on their report, where to go from here. What type of research studies should be considered for the future, addressing this information to your potential readers.
In brief, I believe that this interesting report may be published; at the same time, it is suggested to consider the cited concerns.
Author Response
This is an interesting type of research. With climatic changes which have already arrived; with water limitations in most agricultural areas of the world; and with the increasing necessity of the populations to improve the daily diet with food components and ingredients with outstanding nutrition, nutraceutical and even medical message; then, cacti plants and especially nopal and overall Opuntias genus have become the crop for the 21st century, as it has been classified by international organization.
Therefore, the present report may be very useful to enrich potential options. However, there are important changes before its possible acceptation for publication in Plants.
Objectives - They are acceptable. However, there other equally important objectives which are not covered here; and that they deserve, at least, not to be ignored; on the contrary, authors are suggested to consider. There is one study published in a highly cited journal about changes of the plant in terms of composition, and especially over cladodes, during ripening. Additionally, as important as selecting two different stations of the year, it is equally important, and I do believe that much more important as objective, to study the influence of the geographical region on the composition, especially again, of cladodes; it should be considered here the same cultivar.
Dear Reviwer,
thanks for your valuable advice. As you can see the paper has been remodeled in a clearer form and numerous results have been added.
As you can see, I have better listed the main objectives
Abstract, results and discussion, and conclusions - Authors need to correct the use of cladodes as biological wastes. Cladodes are not wastes at all. Nopal cultivars are not properly used based in their high potential in view of their genetic and agronomical characteristics; but they are not exactly wastes; and it is the same for cladodes. At the end, they are materials not fully used according to the cited potentialities. For the information to authors, cladodes are used in Mexico and Central America, and also in the Mexican - Hispanic populations in the US as very frequent main dishes, and ingredients. And they are called "nopalitos". There is two or three publications from our group, which I should not cite for ethical reasons, about agronomic, reproduction in the field, and composition based on age of the plant ( and consequently age of cladoes) and their influence on such "nopalitos" (cladodes). Therefore, avoid the use of "wastes" or re-difine terms because under such characteristics most cultivar components are "wastes" once they are not fully used.
After reading your suggestions, we increased the study of literature and realized that the definition of "waste" cannot be used for O. ficus-indica cladodes. For this reason, we have eliminated this altered description in every part of the manuscript, highlighting more the uses of cladodes.
Cladodes - Authors are suggested to mention that cladodes contain mucilages and liquid components which, according to the cultivar, etc, they have outstanding nutraceutical and medical components which may be extracted and potentially commercialized.
These suggestions were accepted and included in the work (lines 111-116).
Nopal plant as a bioreactor - There is one or two published studies, based on the performance of the plant under non-adequate environments, which may be genetically modified to insert into them, especailly in the cladodes, the information to express or over-express compounds of pharmaceutical importance; it means, to use the plant as a bioreactor. Authors may use this publication in the Introduction section, as a background.
These suggestions were accepted and included in the work (lines 119-127).
Conclusions - They are really very poor. Authors need to underline, based on their report, where to go from here. What type of research studies should be considered for the future, addressing this information to your potential readers.
In brief, I believe that this interesting report may be published; at the same time, it is suggested to consider the cited concerns.
As suggested, I enriched conclusions and discussion
Reviewer 3 Report
Comments and Suggestions for Authors
The manuscript can NOT be publish in the way it is since a relevant issue of the methods applied was observed.
One of main objectives of the present study was to develop new extraction protocols, able to improve the yield of the fiber obtained. Althought the authors would like to applied a green chemistry method for fiber extraction, they do not take inconsideration the official protocols for fiber quantifications of AOAC, in which there is an specific method that also use green solvents, for samples that present low amount of starch (as in the case of cladodes).
The main problem of the study design was the selected fiber quantificaton method. The water maceration (although was in high temperature and with stirring) only extract the insoluble fraction, without taking into account the soluble fiber fraction (which is aldo of relevant in cladode samples).
From my point of view this is an special matter that limit the accurancy of the obtained results.
On the other hand, it do not make sence the determination bioactive propertise (e.g. Catalase, Superoxide Dismutase, and Glutathione Peroxidase Activities, or cell proliferation) unless a deep sample bioactive compounds characterization was performed (the total phenolic content measure by Folin method is not enought and is not the unspecific method to quantified total phenolic compounds).
Minor comments:
- thermostable alpha-amylase is NOT a protease (pag. 4)
- Data for section 2.5. (Protocols that improve fiber extraction yield do not change the bioavailability of polyphenols) must be included (at least in supplemental material section)
Comments on the Quality of English LanguageMinor mistakes were detected along the manuscript
Author Response

(The authors gave the same response as above.)
